# Mechanical and Microstructure Characterisation of the Hypoeutectic Cast Aluminium Alloy AlSi10Mg Manufactured by the Twin-Roll Casting Process

**Moritz Neuser** [1] **, Mirko Schaper** [1] **and Olexandr Grydin** [2,*]

1 Department of Mechanical Engineering (LWK), Paderborn University, Mersinweg 7, 33100 Paderborn, Germany; neuser@lwk.upb.de (M.N.); schaper@lwk.upb.de (M.S.)
2 Department of Materials Science, Paderborn University, Warburger Straße 100, 33098 Paderborn, Germany
* Correspondence: olexandr.grydin@uni-paderborn.de

**Abstract:** Multi-material designs (MMD) are more frequently used in the automotive industry. Hereby, the combination of different materials, metal sheets, or cast components, is mechanically joined, often by forming joining processes. The cast components mostly used are high-strength, age-hardenable aluminium alloys of the Al–Si system. Here, the low ductility of the AlSi alloys constitutes a challenge because their brittle nature causes cracks during the joining process. However, by using suitable solidification conditions, it is possible to achieve a microstructure with improved mechanical and joining properties. For this study, we used the twin-roll casting process (TRC) with water-cooled rollers to manufacture the hypoeutectic AlSi10Mg for the first time. Hereby, high solidification rates are realisable, which introduces a microstructure that is about four times finer than in the sand casting process. In particular, it is shown that a fine microstructure close to the modification with Na or Sr is achieved by the high solidification rate in the TRC process without using these elements. Based on this, the mechanical properties increase, and especially the ductility is enhanced. Subsequent joining investigations validate the positive influence of a high solidification rate since cracks in joints can be avoided. Finally, a microstructure-property-joint suitability correlation is presented.

**Keywords:** twin-roll casting; aluminium casting alloy; modified microstructure; mechanical properties; AlSi10Mg; mechanical joinability

## 1. Introduction

Lightweight design is a symbiosis of minimising mass and maximising the load-bearing capacity of the component. It is one of the leading leitmotifs in the automotive sector. On the one hand, the goal is to create components that offer weight savings, which leads to resource savings in production and during use, and to contribute to sustainability. On the other hand, the components should be adapted as far as possible to the application of force in terms of geometry and material selection [1,2].

In automotive body design, cast components are used in the spaceframe construction method at the nodes of the chassis that are loaded under pressure [3]. These are usually made of an alloy from the AlSiMg system. These alloys are characterised by suitable mechanical properties, excellent castability, and high corrosion resistance [4]. As different materials have to be joined together in this body design, the use of mechanical joining processes is particularly advantageous. The challenge is to join the aluminium castings without cracks because the brittle material behaviour of the AlSiMg alloys (e.g., AlSi10Mg) leads to the development of cracks during the joining process [5]. In other words, the ductility of the casting material has to be high enough for the joining to be sustainable [6]. One approach to avoiding cracks is to produce cast components that have a very fine fibrous microstructure, which is achieved by a very high solidification rate [7]. As a result, a fine fibrous microstructure can be achieved, which enhances the mechanical properties.

This is also referred to as quench modification by Khan et al. [8] and Hegde et al. [7]. The proportion of flake-like Si decreases progressively with increasing solidification rate, and a finely fibrous Si morphology is formed [8]. The mechanical properties of AlSi alloys depend significantly on the shape of the dendrites [9]. In this way, very small secondary dendrite spacings (DAS) are generated, which in turn leads to an improvement in mechanical properties. The following relationship can be established: The higher the solidification rate to be achieved, the smaller the resulting DAS and the higher the mechanical properties [10,11]. Especially the elongation at fracture and the yield strength are essential criteria that influence the tendency of a component to crack [12]. Casting processes have different solidification rates, which are influenced by the mould material [5]. To create very fine microstructures, it is possible to process aluminium casting alloys such as AlSi10Mg using the twin-roll casting (TRC) process. Here, a high solidification rate is essential for the eutectic Si structure to achieve better elongation at fracture [7].

A further increase in the ductility of the casting can be achieved by subsequent heat treatment, as has been researched in the studies by Jarco [11] and Zhang et al. [12]. Based on these studies, the effect of heat treatment on the mechanical properties and the suitability for joining was investigated in the study of Neuser et al. [13] for the aluminium casting alloy AlSi9. Since this alloy has a higher Si content, comparable to AlSi10Mg, the ductility is also lower than the alloy in the study of Zhang et al. [14]. One possible application is the production of cast semi-finished products made of AlSi alloys, which are produced by means of profiled rolls in the TRC. In particular, applications are conceivable here in which the castings would undergo too high a degree of deformation in conventional forming processes. Using the TRC process, it would be possible to produce castings in a geometry that is already close to the final geometrical contour.

The TRC process combines the primary forming process (casting) with another forming process (rolling), and it differs in the vertical and horizontal process schemes. Thereby, it is divided by the direction of the strip flow, vertical or horizontal. The first variant was used in the present study. In special cases, TRC casters are used, which consist of shells with different diameters [15–17]. In simple terms, such a plant consists of a casting furnace, a nozzle system, a tundish, and a rolling mill. A simplified structure of a TRC plant with two equal shells is shown in Figure 1.

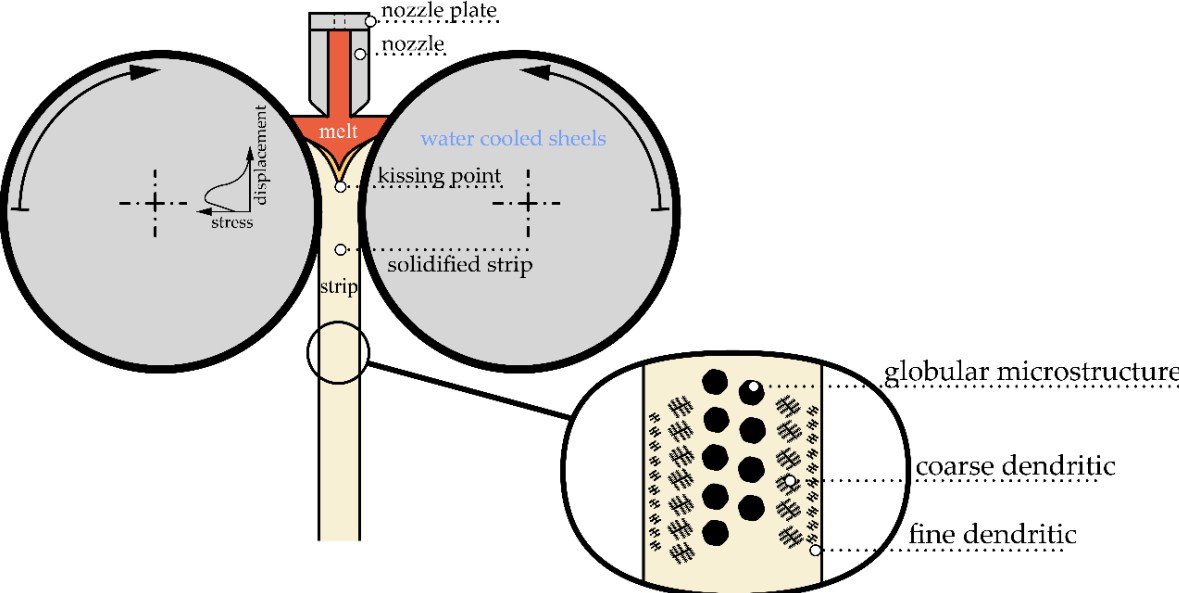

**Figure 1.** Process scheme of vertical TRC and the characteristic microstructure.

Molten aluminium casting alloy, in this case, AlSi10Mg, is fed through the tundish, and the nozzle system feeds into the casting gap. The function of the pre-mounted nozzle

system is to stabilise the melt and enable a homogeneous melt flow into the casting gap. In the casting gap, the melt solidifies on both counter-rotating, internally water-cooled rolling shells. The thickness of the semi-solidified aluminium melt increases as it is moved to the narrowest casting gap by the counter-rotating rolling shells. At the so-called kissing point, the two solidified melt fronts merge. Slightly below the kissing point, the highest forming force occurs, which shapes the strip [18]. A characteristic microstructure is formed, as shown in Figure 1. Due to the rapid solidification, a very fine dendritic microstructure forms in the outer part of the strip. The solidification rate directly at the contact point between the melt and the roller surface is very high [19]. Closer to the centre of the strip, the formation of the microstructure results in a coarser dendritic structure, and a more global microstructure can be seen in the centre of the strip. On the one hand, this is due to the slower solidification rate, and on the other hand, the temperature gradient in the middle of the strip is rather omnidirectional [20]. The thermal conductivity is significantly influenced by the roll material used, which in turn affects the microstructure. For example, the use of copper as a roll material can increase the solidification rate due to the higher thermal conductivity of copper compared to steel [21]. Using the TRC process, strip thicknesses between 0.5 mm and 6.5 mm can be produced [17], whereby solidification rates of about 200 K/s can be achieved [22]. Industrially, wrought aluminium alloys are processed in the TRC Process [17]. Whereas, currently, aluminium casting alloys are not industrially processed. The advantage of the processing of aluminium casting alloys in the TRC process results from the fact that a very fine microstructure can be achieved due to the high solidification rate, which enables a rather high ductility [23]. In addition, the TRC process allows for the production of strips with a lower energy input [15,24]. In the study of Haga et al. [15], a hypereutectic aluminium alloy was processed. Hypoeutectic aluminium alloys have only been successfully processed in the work of Neuser et al. [23]. For this application, the authors processed an AlSi9 alloy on the same caster with steel rolling elements as used in this study. A challenge is the high flowability of hypoeutectic AlSi10Mg as well as the brittle character of this alloy. This issue was also described in the research of Haga et al. [15], but in relation to hypoeutectic AlSi16 and AlSi20, which contain an even higher silicon content.

Therefore, the strip pull-off rate has to be slow enough so that the melt front in the narrowest cross-section of the casting gap is high enough for the production of the strip.

Due to the high solidification rate in the TRC process, not only a fine microstructure can be achieved, but in addition, refining agents can be avoided in the processing of AlSi10Mg. Usually, Na- or Sr-based refining agents are used in hypoeutectic aluminium casting alloys to modify the eutectic microstructure and improve the mechanical properties. The coarse plate-like Si eutectic is thereby modified into a finely dispersed lamellar Si eutectic. Depending on the amount of refining agent used, the eutectic temperature is lowered by about 6–8 °C [7]. The study by Talaat et al. [25] showed that if the solidification rate of the aluminium melt is sufficiently high, the use of refining agents can be avoided. The characteristic flakes of an unmodified AlSi alloy are formed because the aluminium phosphide (AlP) particles provide a nucleus for the Si to accumulate and crystallise on. However, if the undercooling is high enough, the Si will crystallise before it can attach to the AlP [7]. This hypothesis is in accordance with the "restricted-nucleation theory" by Crosley et al. [26]. Typically, refinement is achieved by adding Na or Sr. With a sufficient solidification rate, a Si eutectic is achieved that is almost identical to a refined eutectic [27].

In this study, the successful processing of hypoeutectic AlSi10Mg in the TRC process is shown, and the microstructural, as well as mechanical characterisation, is carried out. Finally, it is discussed how the fine-adjusted microstructure positively influences the mechanical joining process by reducing the susceptibility to cracking. The aim of this study is to evaluate a microstructure-property-joinability correlation.

## 2. Materials and Methods

### 2.1. Aluminium Casting Alloy EN AC-AlSi10Mg and Joining Partner Dual-Phase Steel HCT590x

The hypoeutectic, age-hardenable aluminium casting alloy EN AC-AlSi10Mg (i.e., European Norm—aluminium cast product) is used in this study. AlSi10Mg is suitable for casting and has favourable mechanical properties (e.g., high strength as well as hardness and dynamic load capacity), which can be further enhanced by additional precipitation hardening. The chemical composition of the alloy is determined with the optical emission spectroscope (OES) of the Bruker model Q4 Tasman.

In addition, the dual-phase steel HCT590x is used as the joining partner on the punch side during the joining examinations. HCT590x is mostly used in crash-relevant parts of vehicle bodies because of its high energy absorption.

### 2.2. Twin Roll Casting Process

A vertical casting-rolling machine was used to process the AlSi10Mg [28]. The casting machine consists of two fluid-cooled rolling shells made of X38CrMoV5-3 with a diameter of 370 mm and a width of 200 mm. A glycol–water mixture is used as a cooling medium. A cooling medium flow rate of 2.4 m³/h was applied for the investigations. The roll temperature at the beginning of the trial was 15 °C. A resistance furnace with a bottom outlet and a power of 2.8 kW is used to melt the AlSi10Mg ingots. The melt is poured at a temperature of 635 °C via the tundish, which is equipped with a nozzle plate, into a distribution nozzle. At the opening of the nozzle, the melt had a maximal temperature of 589 °C. The nozzle plate and the nozzle were milled from calcium silicate plates. The remaining dimensions are shown in Figure 2, which contribute significantly to the formation of the later strip. The melt introduced through the nozzle then enters the 0.9 mm-wide rolling gap, where a quasi-simultaneous primary forming as well as forming process is performed.

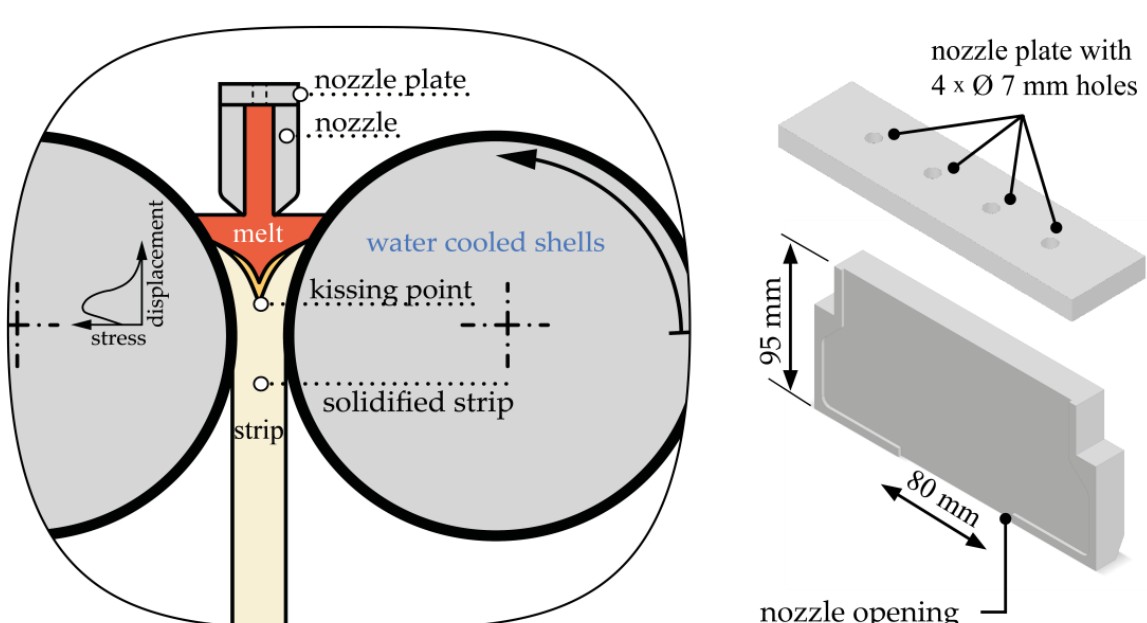

**Figure 2.** TRC process and the geometry of the nozzle plate as well as the nozzle.

The total mean roll separating force (RSF) during the trial was 230 kN with a standard deviation of 40 kN, which was recorded at a sampling rate of 100 Hz. At the beginning of the strip, an average RSF of 272 kN was measured. After the steady state of the TRC process was reached, a mean RFS of 166 kN with a deviation of 19 kN was measured. A strip of hypoeutectic AlSi10Mg approximately 11 m long and 1.9 mm thick was produced using these parameters.

### 2.3. Microscopic Characterisation and Determination of DAS

To carry out the microstructural characterisation, the samples were embedded using the conductive warm embedding (15 min at 180 °C) medium WEM REM from Cloeren Technology (Germany). Subsequently, the samples were ground in steps up to a grit size of 2500 with silicon carbide grinding paper and finally polished for 18 h with an ultrasonic polishing machine using the silica polishing suspension (pH value: 9.5–10, grit size: 50 nm) from Cloeren Technology.

For the following examinations, samples were examined under three different conditions: (1) as-cast, (2) artificially aged at 120 °C, and (3) at 170 °C (Table 1). The temperature for the artificial ageing was selected to achieve two purposes: to generate a microstructure with very ductile properties and to realise the maximum achievable strength. For this purpose, various heat treatment conditions of AlSi10Mg were evaluated with regard to these two aspects in preliminary investigations. The results of these preliminary investigations are the two temperature levels used here. In the following, the specimens corresponding to the artificial ageing condition at 120 °C are referred to as ht1 and at 170 °C as ht2. All three specimen conditions were used throughout all examination methods.

**Table 1.** Heat treatment of specimens.

| Conditions | Solution Annealed Temperature in °C | Soaking Time in h | Artificially Aged Temperature in °C | Soaking Time in h |
|---|---|---|---|---|
| as-cast | - | | - | |
| ht1 | 525 °C | 4 h | 120 °C | 4.5 h |
| ht2 | 525 °C | 4 h | 170 °C | 4.5 h |

Due to the directional solidification caused by the internally cooled rollers, the AlSi10Mg solidifies in a dendritic formation. Based on the secondary dendrites, the characteristic dendrite arm spacing (DAS) for each solidification rate can be determined. For this purpose, the length of a dendrite stem consisting of at least 5 individual dendrites is measured, as detailed and illustrated in the guideline P220 [29]. Knowing the number of dendrites, the DAS can be determined by Equation (1). Furthermore, it is possible to calculate the solidification time using Equation (2). It requires knowledge of the DAS as well as a material-specific characteristic parameter, which in the case of AlSi10Mg has a value of 10.4 [29]. Finally, the solidification rate can be calculated. using Equation (3). Here, the BDG guideline P220 [29] specifies a solidification interval of 45 K for the AlSi10Mg alloy.

$$DAS = \frac{x}{m-1} \tag{1}$$

$$t_f = \left(\frac{DAS}{k}\right)^3 \tag{2}$$

$$SR = \frac{T_{a-e}}{t_f} \tag{3}$$

where:
  $DAS$ = secondary dendrite arm spacing
  $x$ = length of the dendritic stem
  $m$ = number of dendrites
  $k$ = material-specific parameter
  $SR$ = Solidification rate
  $T_{a-e}$ = solidification interval
  $t_f$ = solidification time

To analyse the grain structure of AlSi10Mg produced by the TRC process, electron backscatter detector (EBSD) images were taken using the Zeiss Ultra Plus scanning electron

microscope (SEM) with an EBSD from AMETEK. A magnification of 500, a working distance of about 13.5 mm, an accelerating voltage of 20 kV, and a step size of 0.2 were selected for the EBSD images on the SEM. The graphic of the EBSD analysis shows an area of 175 × 185 μm. These EBSD measurements are all uncleaned. In addition, an energy dispersive X-ray spectroscopy (EDS) analysis was carried out on the same section and parameters of the EBSD scan.

*2.4. Mechanical Testing*

The mechanical tests were carried out with tensile specimens based on DIN EN ISO 6892-1 and Brinell hardness tests according to DIN EN ISO 6506-1.

The geometry of the tensile specimens is shown in Figure 3. The tensile tests are carried out using the tensile testing machine MTS Table Top with a testing rate of 1.5 mm/min and a clamping pressure of 8 MPa. At least 9 specimens were tested per condition. The force and strain were recorded at a sampling rate of 1024 Hz.

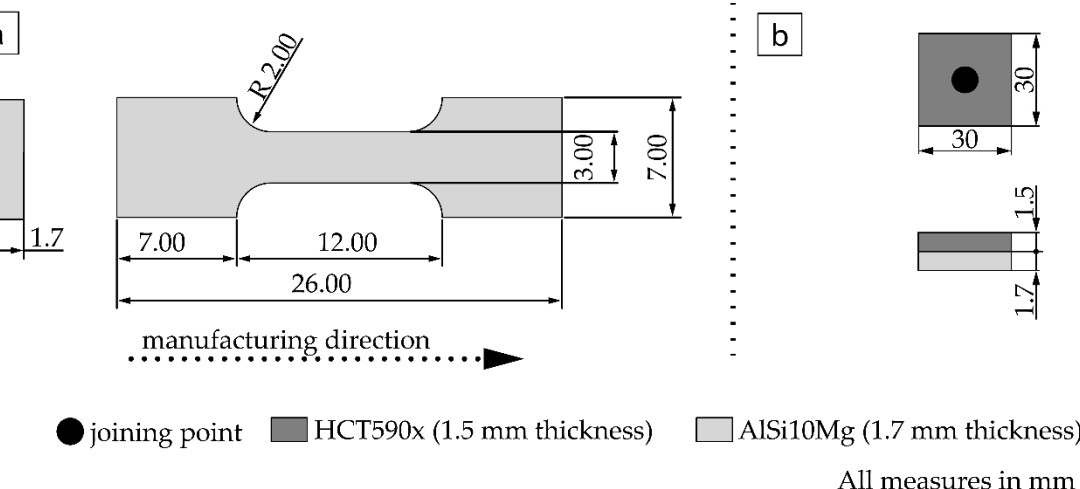

**Figure 3.** (**a**) Geometry of the tensile test specimen and (**b**) geometry of the joint sampling specimen according to DVS (German Welding Society) guideline 3480-1 [30].

The hardness tests were carried out on specimens prepared with sandpaper with a grit size of 1200 before measuring. A load of 612.9 N and a tungsten carbide ball with a diameter of 2.5 mm were used for the test. Following DIN EN ISO 6506-1, the test force was applied for 10 to 15 s. Here, at least five measurements per condition were carried out for this application.

To validate the suitability for joining, so-called sampling specimens were prepared (see Figure 3). The material combination HCT590x/AlSi10Mg was used for this application. Self-piercing riveting (SPR) was used as the joining technique. On the punch side, the steel sheet (HCT590x) and, on the die side, the cast aluminium castings were used. The SPR system Tox-TE-x was utilised to join the joints using the die type FM 100 2213 and the rivet C 5X5.5 H4. To adjust the joining process, cross-sections were made, and quality-relevant parameters such as residual bottom thickness, rivet head end position, and interlock were determined.

## 3. Results and Discussion

### 3.1. Chemical Composition and Microstructure

As part of this study, a chemical analysis of the AlSi10Mg test material was carried out after the TRC process. The result is listed in Table 2 and corresponds to the specifications of the DIN EN 1706 standard; e.g., the Si content of 10.18 wt% is within the specified range. Concerning the subsequent heat treatment, a content of 0.386 wt% Mg is important to enable precipitation hardening by the formation of $Mg_2Si$ precipitates. Furthermore, the

AlSi10Mg used does not contain significant amounts of refining agents such as Na or Sr. An amount of 0.02 wt% (Sr), or rather 0.01 wt% (Na), would be necessary for a refining effect [31]. The refining effect shown later is achieved by the high solidification rate.

**Table 2.** Chemical composition of AlSi10Mg after processing via TRC determined with OES.

| AlSi10Mg | | | | | | | | |
|---|---|---|---|---|---|---|---|---|
| Elements | Al | Si | Mg | Fe | Cu | Na | Sr | Others |
| Mean valuein wt% | 88.73 | 10.18 | 0.386 | 0.122 | 0.0028 | 0.0001 | 0.00005 | 0.5791 |
| Standard deviation | 0.05 | 0.054 | 0.002 | 0.001 | <0.000 | <0.000 | <0.000 | |

### 3.2. Microscopic Analyses

Figure 4 shows the difference in the microstructure of the AlSi10Mg manufactured by sand casting (a) and TRC (b). As a comparison, we use the results of our previous study by Neuser et al. [12]. In this study, the experimental design of the sand casting trials is also explained in detail. Remarkably, the AlSi10Mg processed in sand casting (a) shows the typical microstructure, i.e., a coarse plate-like Si eutectic, of a non-refined, hypoeutectic AlSi alloy. Due to the coarse and plate-like structure of the eutectic, ductility is low [32]. In sand casting, only moderate solidification rates of about 4.4 K/s are achieved with a component thickness of 2 mm [12]. In comparison, the microstructure shown in (b) is clearly different due to the processing via TRC. The dendritic structure of the α-aluminium is visible. In addition, there is no plate-like Si eutectic here, but a very finely separated, lamellar, and partially spherical Si eutectic. As a result of the process-specific solidification rates, there is a difference in the microstructure, which is also shown in the studies of Talaat et al. [25] and Fredriksson et al. [27]. However, as mentioned above, fast solidification suppresses the primary formation of a plate-like Si eutectic, as the Si cannot solidify on the AlP particles but solidifies without forming a plate-like morphology [25,27]. The formation of plate-like Si results in a decrease in ductility [32].

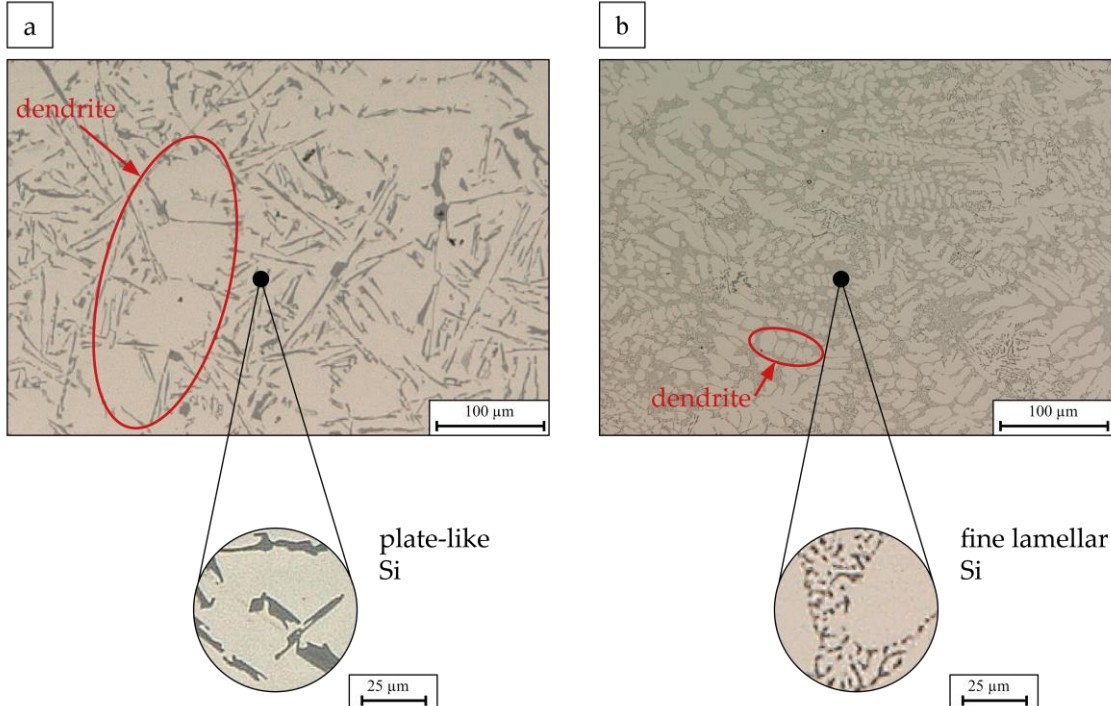

**Figure 4.** Comparison of microstructure; (**a**) microstructure after sand casting without modification element; (**b**) microstructure after TRC without modification element.

The results of the DAS measurement across three-strip segments are shown in Table 3. In Figure 4, exemplary dendrite stems are marked in red, which can be used for the DAS measurement. In the beginning phase of the process, the DAS is 10.24 μm. As the solidification front in the casting, the gap is not yet uniform, there are large differences between the edges of the strip (DAS 11.4 μm and 8.4 μm, respectively) and the middle section of the first strip segment (10.9 μm). To calculate the solidification rate based on Equations (1) to (3), the evaluation of the DAS measurement of the AlSi10Mg processed via TRC is used. Thus, the solidification rates in the first strip section vary between 33.84 K/s and 84.97 K/s. This is due to a relatively high melt quantity accumulated in the roll gap, resulting in a high RSF at the beginning. If, however, the TRC process reached its steady state, recognisable by the constant DAS and the resulting solidification rate, the casting rate could be increased and, as a result, the RSF would decrease.

**Table 3.** Results of DAS measurement and the associated calculated solidification rates. Values are calculated according to guideline P220 [29].

| Section of the Strip | DAS in μm | DAS Standard Deviation in μm | Solidification Time $t_f$ in s | Solidification Rate $SR$ in K/s |
|---|---|---|---|---|
| begin | 10.24 | 3.06 | 0.96 | 47.07 |
| middle | 6.54 | 0.89 | 0.25 | 181.24 |
| end | 6.64 | 0.95 | 0.26 | 172.65 |

The solidification rate at the beginning of the trial was 47.07 K/s and 181.24 K/s in the middle of the strip. Due to the constant DAS indicated by the low standard deviation and the constant solidification rate, a steady-adjusted process can be assumed from the middle to the end of the strip. As the roller shells increase in temperature during the trial, the solidification rate also decreases slightly. Compared to sand casting, the solidification rate in TRC (steady state) using steel shells is 41 times higher [12]. However, the DAS is 15.8 μm in comparable sand-casting specimens. Hence, the significantly higher solidification rate (4.42 K/s [12] to 181.24 K/s) leads to a smaller DAS, which also corresponds to the common relationship between the solidification rate and DAS [33]. In the research of Kong et al. [34], different solidification scenarios were investigated for an AlCu alloy. They also determined DAS similar to ours using the AlCu alloy in a cooled sand mould.

The study by Talaat et al. [25] shows that the inter-particle spacing of unmodified and modified melts is about 1 μm [25] at a solidification rate of just below 100 K/s. This indicates that the eutectic crystallises at a similar rate in both cases. Consequently, the solidification rate has a primary influence on the eutectic structure, beyond the use of refining agents [25]. In general, the addition of Na or Sr as a refining agent will result in a decrease in the eutectic temperature. As a result, the nucleation of Si on AlP particles is inferior, leading to the formation of a fibrous eutectic [32]. It can be assumed that, due to the rapid solidification rate, there is a high degree of undercooling, and therefore the Si eutectic nucleates faster than at equilibrium eutectic temperature with the reaction of Si with AlP. This means that the Si crystallises faster than it can form through a reaction with AlP needle-like particles.

To further examine the AlSi10Mg, EBSD, and EDX analyses were performed (Figure 5). The images show a section in the middle of a band. As already identified in Figure 4, a globular solidification morphology in the centre of the strip is particularly well visible in the grain highlighted in red. Crystal orientations are evident, indicating that solidification has proceeded homogeneously in all spatial directions. Microscope images from the edge section, i.e., the area that had direct contact with the shells, show a clear dendritic solidification.

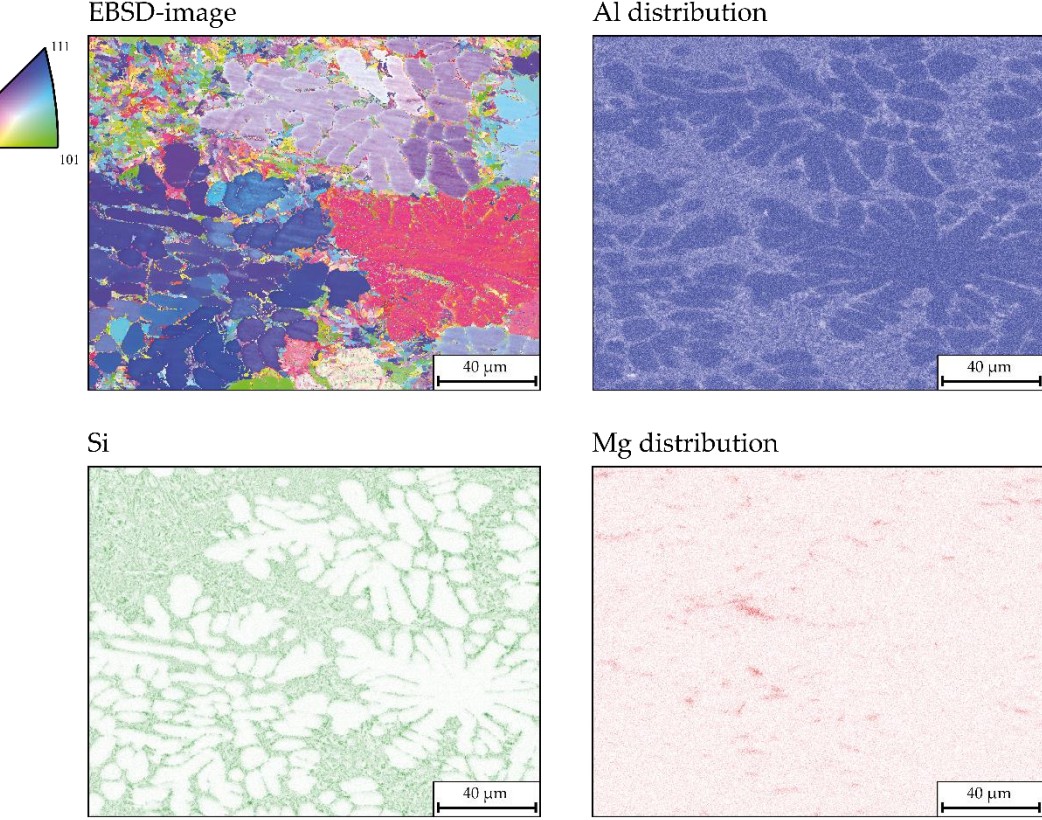

**Figure 5.** EBSD and EDX analyses of AlSi10Mg manufactured by TRC.

In addition, EDX analyses were carried out for the main elements Al, Si, and Mg. As expected, Al is found in the α-phase of the dendrites as well as in the eutectic. The EDX image of the Si particles illustrates the already discussed refinement-like morphology of the Si eutectic due to the high solidification rate without the use of refining agents. A large part of the Si eutectic is present as a lamellar to spherical morphology.

### 3.3. Mechanical Properties

Hardness measurements of the AlSi10Mg processed in the TRC process (Table 4) show that—as expected—the precipitation hardening increased the hardness of the aluminium. A comparison of the as-cast specimens and the artificially aged specimens at 170 °C showed an increase in hardness of 41.8%. Due to the diffusion processes not being entirely carried out at an artificial ageing temperature of 120 °C, the hardness is reduced by 3.9%. Compared to the previous study [12], which was based on specimens produced by the sand casting process, the measured hardness values are at a similar level. In the T6 condition (i.e., artificial ageing at 170 °C), the average hardness could be increased by 1.2 % for the TRC-processed specimens in contrast to the sand-cast specimens [12].

**Table 4.** Hardness values of AlSi10Mg manufactured in the TRC process compared in three conditions.

| Conditions | As-Cast | ht1 = Solution Annealed (525 °C) and Artificially Aged (120 °C) | ht2 = Solution Annealed (525 °C) and Artificially Aged (170 °C) |
|---|---|---|---|
| Hardness in HBW 2.5/62.5 | 73.2 | 70.4 | 107.3 |
| Standard deviation in HBW | 3 | 2 | 2 |

The mechanical properties of the specimens produced under different conditions in the TRC process are shown in Figure 6. There was a 6.5% difference in tensile strength between the as-cast and the ht1 condition. However, the elongation at the fracture could be increased by 2.7 times. In comparison, an increase in elongation at the fracture of 59.8% was determined between the as-cast and the ht2 conditions, and the tensile strength increased by 23.1%. The highest possible elongation at fracture should be aimed for in the subsequent mechanical joining. According to DVS guideline 3420, a suitable elongation at fracture for clinching is at least 12%; for the ht1 condition, the threshold value is achieved [35]. Furthermore, a yield strength ratio, the quotient of yield strength and tensile strength, of less than 0.7 is recommended for clinching suitability. In the as-cast and under the ht1 condition, the limit value is achieved at 0.49 and 0.48, respectively. Under the ht2 condition, however, the limit value of 0.75 is slightly exceeded. Thus, material condition ht1 can be classified as well-suited for clinching according to DVS guideline 3420. In contrast, material condition ht2 can only be described as partially suitable for clinching (cf. [35]).

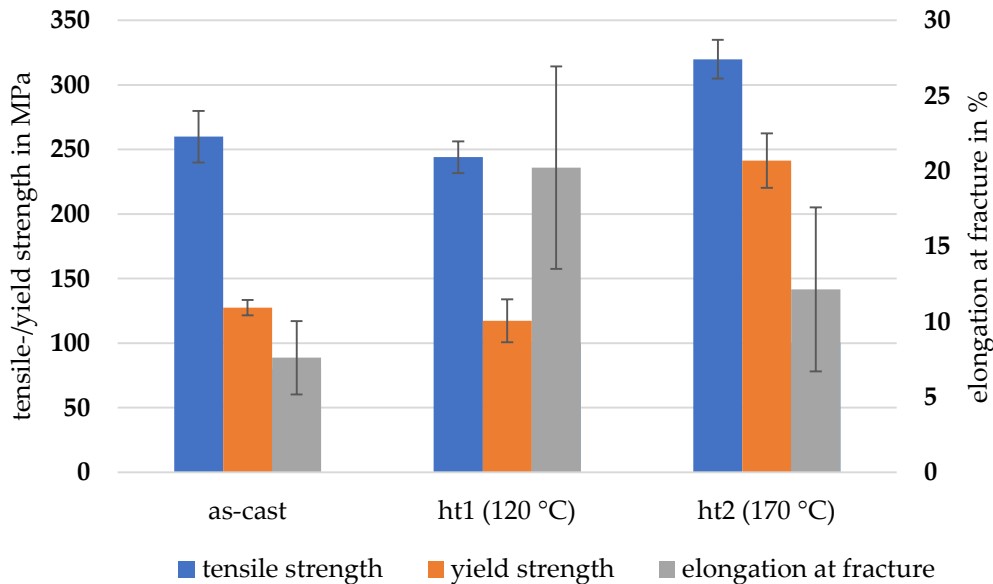

**Figure 6.** Mechanical properties of AlSi10Mg manufactured in the TRC process as a result of various material conditions.

As for the microstructure, differences observed in the mechanical properties of samples produced in the sand casting and TRC processes can be seen. The mechanically characteristic values from the study by Neuser et al. [12] are used as comparative data. Within this study, plates consisting of AlSi10Mg were cast in different thicknesses using the sand casting technique and mechanically characterised. In particular, the mechanical properties of the 2 mm-thick plates served as a reference value (Table 5). It is evident that the mechanical properties of tensile strength and elongation at fracture increase due to the increase in solidification rate, especially concerning the elongation at fracture, however, there is no linear correlation between solidification rate and mechanical properties. The higher solidification rate compared to sand casting results in a finer microstructure, which generates more grain boundaries and subsequently increases ductility. An increase in the solidification rate from 4.42 K/s to 181 K/s leads to an enhancement of the elongation at fracture of 85% and 48%, while the DAS is reduced by a factor of 2.42 [12].

**Table 5.** Comparison of mechanical data for sand casting and TRC specimens. Data for sand casting is based on the study by Neuser et al. [36].

| Process | Sand Casting | TRC | Increase |
|---|---|---|---|
| Tensile strength in MPa (as-cast) | 223 | 259 | 16% |
| Tensile strength in MPa (T6/ht2) | 286 | 319 | 11% |
| Elongation at fracture in % (as-cast) | 4.1 | 7.6 | 85% |
| Elongation at fracture in % (T6/ht2) | 8.2 | 12.1 | 48% |

*3.4. Joining Suitability by Using SPR*

Within the scope of the investigations, SPR samples were prepared, and the quality-relevant parameters RE (rivet head end position), IL (interlock), and MBT (minimum bottom thickness at the base of the rivet) were determined. Figure 7 shows a cross-section of an SPR joint representing each of the three material conditions. The visual examination of the entire cross-section shows no intrinsic cracks in the joint. This aspect is related to the comparatively small DAS values of 6.54 µm and 6.64 µm, respectively, especially compared to sand cast specimens [12], the increase in ductility due to the high solidification rate, and follows the mechanical properties discussed previously. Although the casting undergoes high deformation due to the joining process, the increased ductility is sufficient to ensure that no intrinsic cracks occur. In contrast, the solidification rate to be achieved is too low, and the resulting elongation at fracture is accordingly insufficient, not in accordance with DVS Guideline 3420 [35], and cracks occur, as has already been shown in the study by Neuser et al. [12]. The results of the sampling examination of the three conditions processed in TRC are listed in Table 6.

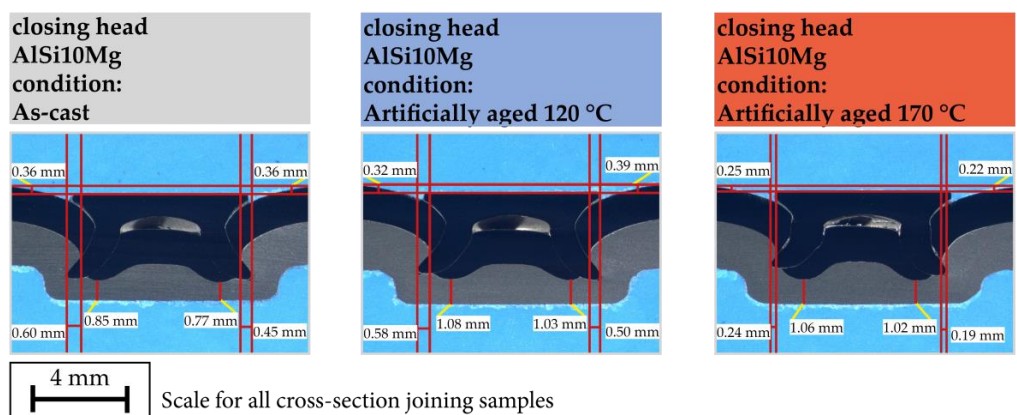

**Figure 7.** Joining sampling of SPR joint consists of HCT590X (punch-sided) and three conditions of AlSi10Mg (die-sided): as-cast, ht1, and ht2.

**Table 6.** Quality-relevant parameters of SPR joints. Abbreviations: RE (rivet head end position); IL (interlock); and MBT (minimum bottom thickness at the base of the rivet).

| Mean Values of SPR Joints | | | |
|---|---|---|---|
| Condition | RE in mm | IL in mm | MBT in mm |
| as-cast | 0.38 | 0.54 | 0.84 |
| ht1 (120 °C) | 0.35 | 0.43 | 1.00 |
| ht2 (170 °C) | 0.23 | 0.29 | 0.84 |

Table 6 lists the determined quality-relevant characteristic values of the SPR joints. Compared to the joining quality-relevant characteristic values determined in the study by Neuser et al. [37], the hypoeutectic, naturally hardened AlSi9 was processed in the TRC process. For all three parameters relevant to joining quality, comparable values were obtained in the present study. In particular, the rivet head positions of 0.38 mm and 0.35 mm were similar to AlSi9. However, a value of only 0.23 mm was reached in the ht2 condition. This means that the rivet penetrates deeper into the material, but in the ht2 condition, there is only an interlock of 0.29 mm. Compared to the other two states and the study [37], this value is relatively low, which ultimately results in a lower form fit of the joint. The high yield strength and low elongation at fracture of the condition ht2 result in a negative influence on the material flow during the joining process, so that the interlock can only be formed to a limited extent in contrast to the conditions as-cast and ht1.

As detailed above, the high solidification rate leads to increased mechanical properties, which in turn significantly determine the joining suitability of the material. To evaluate the increased joining suitability due to a finer microstructure, the closing heads of SPR joints were examined for cracks. For this purpose, Figure 8 shows a large number of closing heads for the three material conditions. Although a yield tensile strength ratio of 0.49 could be determined for the as-cast condition, cracks were detected in all locking heads. An increase in yield strength was achieved compared to sand casting due to the high solidification rate, though the elongation at fracture of 7.6% is insufficient to allow crack-free joining.

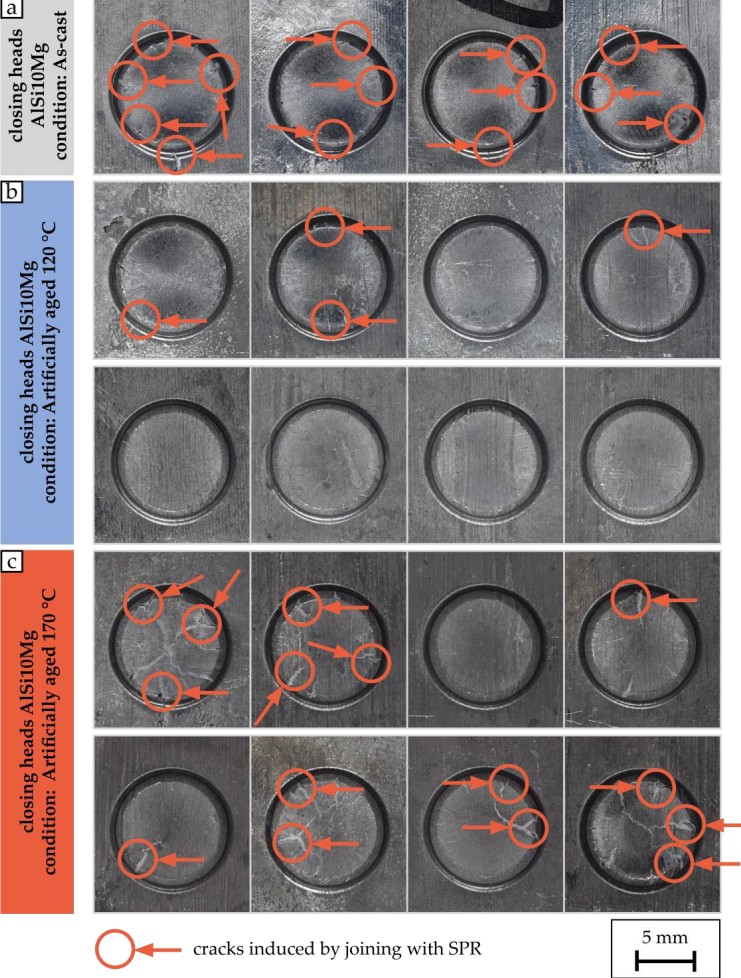

**Figure 8.** Closing heads of SPR joints consisting of AlSi10Mg manufactured in TRC. (**a**) AlSi10Mg in as-cast, (**b**) AlSi10Mg in ht1-condition (120 °C), and (**c**) AlSi10Mg in ht2 condition (170 °C).

With a similar yield strength ratio of 0.48, crack initiation can only be detected in occasional cases in the specimens heat-treated under the ht1 (120 °C) condition. Due to the increase in elongation at fracture achieved by artificial ageing at 120 °C, a crack-free joint can be produced in the majority of cases. It should be noted here that the four determined cracks have a rather short crack length of less than 1 mm.

In contrast, a clear difference between the heat-treated samples under the ht2 (170 °C) condition is evident. Due to the disproportionate increase in yield strength with a comparatively low elongation at fracture, there is likely only limited suitability for joining with the previously calculated yield tensile strength ratio of 0.75. This can be confirmed by the examination of the closing heads in Figure 8. Not only has the number of cracks increased considerably compared to the as-cast condition, but the cracks are also longer and deeper. Furthermore, due to the low ductility, the material separated at the edge of the locking head.

## 4. Conclusions

Within the scope of the present examination, the hypoeutectic aluminium casting alloy AlSi10Mg was successfully processed in the twin-roll casting (TRC) process. In particular, the brittle character of this aluminium casting alloy, which is due to the high Si content of 10.18 wt.%, poses a challenge for the TRC process. From the mechanical, microstructural, and joining investigations, the following conclusions can be drawn regarding the material processed via TRC:

Due to the high solidification rate of 181 K/s, a cast strip with a fine microstructure (i.e., DAS is on average 6.54 μm to 6.65 μm) can be manufactured.

Since a solidification rate of between 172.65 K/s and 181.24 K/s is achieved in the TRC process, it is possible to prevent crystallisation of the Si on aluminium phosphide (AlP), resulting in a very fibrous to spherical morphology of the Si, without the use of refining agents.

The mechanical properties prove that the higher the solidification rates, the higher the associated mechanical properties. Particularly, the fine microstructure and the associated increase in grain boundary surface result in significantly improved elongation at fracture compared to sand casting. This characteristic value is particularly relevant to the suitability of aluminium casting alloys to be joined without cracking, as demonstrated by the analysis of the locking heads.

In addition, it is necessary to carry out a precipitation hardening process at an artificial ageing temperature of 120 °C to obtain an almost crack-free joint of AlSi10Mg fabricated in the TRC process. Furthermore, the measurement of the quality-relevant characteristic values indicates that this heat treatment strategy results in suitable joinability for aluminium casting alloys. In this study, it was determined for the aluminium casting alloy AlSi10Mg that there is a correlation between the microstructure, the mechanical properties, and the resulting suitability for mechanical joining. By means of specifically applied high solidification rates, a fine microstructure is achieved, which enhances the mechanical properties, in particular the elongation at fracture and the yield strength. These are essential parameters to realise a suitable joining suitability.

**Author Contributions:** Conceptualization, M.N. and O.G.; methodology, M.N.; software, M.N.; validation, M.N.; formal analysis, M.N.; investigation, M.N.; resources, M.S.; data curation, M.N.; writing—original draft preparation, M.N.; writing—review and editing, O.G.; visualization, M.N.; supervision, O.G. and M.S.; project administration, M.S.; funding acquisition, M.S. All authors have read and agreed to the published version of the manuscript.

**Funding:** This research was funded by the Deutsche Forschungsgemeinschaft (DFG, German Research Foundation)—TRR 285/2—Project ID 418701707. The authors thank the German Research Foundation for their organisational and financial support.

**Data Availability Statement:** The data will be provided by the corresponding author upon request.

**Acknowledgments:** Furthermore, the authors would like to take this opportunity to thank Hendrik Bücker, Paderborn University, and Martin Lauth, as well as Oliver Hornberger, Materials Science at Paderborn University, for their support in the preparation and conduction of casting trials, as well as Ina Stratmann, Institute for Rail Vehicles and Transport Systems at RWTH Aachen University, Tanja Stratman, Royal Netherlands Institute for Sea Research (NIOZ), and Jan Tobias Krüger, Materials Science at Paderborn University, for their support in revision.

**Conflicts of Interest:** The authors declare no conflict of interest.

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
