# Peer review of "Mechanical and Microstructure Characterisation of the Hypoeutectic Cast Aluminium Alloy AlSi10Mg Manufactured by the Twin-Roll Casting Process"

_jmmp, doi:10.3390/jmmp7040132_

Round 1

Reviewer 1 Report

The manuscript entitled "Mechanical and microstructure characterisation of the hypoeutectic cast aluminium alloy AlSi10Mg manufactured by the twin-roll casting process" introduced using twin-roll casting process (TRC)  with water-cooled rollers to manufacture the hypoeutectic AlSi10Mg helps achieving a microstructure that is around four times finer than in the traditional san casting process. Mechanical and material characterization are used as evidence to show the benefit from the finer microstructure.

1. Abstract should polish more to show the potential applications.

2. Section 2.2, the standard deviation of RSF is around 20%, can the author elucidates more on this? why this acceptable?

3. Table 2, the accuracy is different for different elements. Can the authors formate this table?

4. Based on Figure 6, at 120 C, the elongations possesses the highest improvement, however, when the temperature increased to 170C, the elongation decreases. Is it due to the flow stress change? Normally, the elongation is a property related to fracture. The temperature difference between 170C and 120C is that much. Can the authors elucidate more on this point and explains why it's reasonable to make the comparison between 120 C and 170C? 

English is find.

Author Response

Dear Reviewer,

Thank you for your time and efforts to improve our paper. Please, finde our responses attached in the PDF file.

Best regards

Reviewer 2 Report

This paper is on twin roller casting of Al-Si-Mg alloys to produce metal sheets. The microstructure of cast components and their mechanical property were studied. The relatively high cooling rare resulted in fine structure of the hardenable aluminum alloys of the Al-Si-Mg system. Improved ductility of the Al-Si-Mg alloys was achieved by using the twin roller casting process. The twin-roll casting process (TRC) with water-cooled rollers enhanced the solidification and high cooling rates are realized. The microstructure is about four times finer than that obtained in the sand casting process. It is also shown that the fine microstructure allows the mechanical properties to increase. Especially the ductility is enhanced. This is a nice experimental work. Data support conclusions. It is acceptable for publishing.

Author Response

Dear Reviewer,

thank you very much for your time and efforts in order to improve our paper. We were very pleased to receive your comment.

Best regards

Reviewer 3 Report

1.     Table 2. With what accuracy were the chemical composition measurements carried out?In section 2. Materials and Methods, the authors did not indicate how the EBSD and EDX analyses were performed.

2.     The authors pay a lot of attention to the results of DAS measurement on three-strip segments. It would be useful to indicate DAS with arrows in Figures 4 and 5.

3.     Line 201 - What does "DVS guideline" mean? Many readers do not know what this abbreviation means.

4.     Why did the authors choose an indenter with a diameter of 2.5 mm and a load of 612.5 N to measure Brinell hardness? The typical test uses a 10 mm (0.39 in) diameter steel ball as an indenter with a 3,000 kgf (29.42 kN; 6,614 lbf) force. Is it possible to compare the results here with other measurements of the hardness of metals and alloys?

5.     Line 301 - The authors write "This aspect is associated with the fine microstructure and the increase in ductility due to the high solidification rate and is following the mechanical properties discussed before". Here it is necessary to give a more detailed explanation of what "fine microstructure" we are talking about.

Minor editing of English language required

Author Response

(The authors gave the same response as above.)
